# Adapting Contrastive Language-Image Pretrained (CLIP) Models for Out-of-Distribution Detection

**Nikolas Adaloglou**                                            *adaloglo@hhu.he*
*Heinrich Heine University, Duesseldorf*

**Felix Michels**                                              *felix.michels@hhu.de*
*Heinrich Heine University, Duesseldorf*

**Tim Kaiser**                                                  *tikai103@hhu.de*
*Heinrich Heine University, Duesseldorf*

**Markus Kollman**                                          *markus.kollmann@hhu.de*
*Heinrich Heine University, Duesseldorf*

**Reviewed on OpenReview:** *https: // openreview. net/ forum? id= YCgX7sJRF1*

## Abstract

We present a comprehensive experimental study on pre-trained feature extractors for visual out-of-distribution (OOD) detection, focusing on leveraging contrastive language-image pre-trained (CLIP) models. Without fine-tuning on the training data, we are able to establish a positive correlation ($R^2 \geq 0.92$) between in-distribution classification and unsupervised OOD detection for CLIP models in 4 benchmarks. We further propose a new simple and scalable method called *pseudo-label probing* (PLP) that adapts vision-language models for OOD detection. Given a set of label names of the training set, PLP trains a linear layer using the pseudo-labels derived from the text encoder of CLIP. Intriguingly, we show that without modifying the weights of CLIP or training additional image/text encoders (i) PLP outperforms the previous state-of-the-art on all 5 large-scale benchmarks based on ImageNet, specifically by an average AUROC gain of 3.4% using the largest CLIP model (ViT-G), (ii) linear probing outperforms fine-tuning by large margins for CLIP architectures (i.e. CLIP ViT-H achieves a mean gain of 7.3% AUROC on average on all ImageNet-based benchmarks), and (iii) billion-parameter CLIP models still fail at detecting feature-based adversarially manipulated OOD images. The code is available at `https://github.com/HHU-MMBS/plp-official-tmlr2024`.

## 1 Introduction

Transferring the representations of pretrained vision models has improved the performance on a plethora of image recognition tasks (Yosinski et al., 2014; Tan et al., 2018; Park et al., 2023; Adaloglou et al., 2023). To date, these models are trained with various types of supervision, which accelerates training convergence compared to random initialization (He et al., 2019). Examples include self-supervision (Chen et al., 2020b), natural language supervision (Radford et al., 2021), weakly-supervised learning (Mahajan et al., 2018), or standard supervised learning. Concurrently, Dosovitskiy et al. (2021) have established vision transformers (ViTs), along with an enormous number of variants (Liu et al., 2021; Touvron et al., 2021; Beyer et al., 2023), as a suitable architecture for training large-scale models in the visual domain (Dehghani et al., 2023).

Nevertheless, the applicability of the learned features of pretrained models is crucial and non-trivial, especially for unsupervised downstream tasks (Bommasani et al., 2021; Adaloglou et al., 2023). This work centers on adapting pretrained feature extractors for various visual OOD detection setups, focusing on contrastive language-image pretraining (*CLIP*) models.

The task of OOD, novelty, or anomaly detection aims at identifying whether a given test sample is drawn from the *in-distribution* (the training set) or an alternative distribution, known as the *out-distribution*. Accurate detection of anomalies is indispensable for real-world applications to ensure safety during deployment (Amodei et al., 2016; Ren et al., 2019). The detected unfamiliar samples can be processed separately, possibly with a human expert in the loop, rather than making a potentially uncalibrated prediction (Guo et al., 2017). Despite significant advances in deep learning, neural networks tend to generate systematic errors for test examples far from the training set (Nguyen et al., 2015) or assign higher likelihoods to OOD samples compared to in-distribution samples (Nalisnick et al., 2019).

Recent studies have established a firm connection between the training data distribution accuracy and OOD generalization (Hendrycks et al., 2021a; Wenzel et al., 2022; Dehghani et al., 2023). A similar connection has been identified for supervised OOD detection when in-distribution training labels are available for training or fine-tuning. Supervised training leads to intermediate representations that likely form tight label-related clusters (Fort et al., 2021). An ideal representation for OOD detection should capture semantic properties, such as the pose and shape of an object, while remaining sensitive to the properties of the imaging process (e.g., lighting, resolution) (Winkens et al., 2020).

A suitable choice of visual feature representations is critical for detecting anomalies. However, learning informative representations for unsupervised OOD detection (Tack et al., 2020; Sehwag et al., 2021; Rafiee et al., 2022), where no in-distribution labels are available, is a challenging and active research area (Cohen et al., 2023). Unsupervised methods often adopt self-supervision to learn the in-distribution features by defining pretext tasks such as rotation prediction (Gidaris et al., 2018). A major milestone in visual representation learning was reached by Chen et al. (2020b) with the development of contrastive and non-contrastive visual self-supervised methods (Caron et al., 2021). Recently, CLIP has enabled learning from vast amounts of raw text (Radford et al., 2021). Vision-language models offer the unprecedented advantage of zero-shot image classification by leveraging the label names of the downstream dataset. Nonetheless, adapting CLIP models is not straightforward, even in the supervised scenario. For instance, Pham et al. (2023) showed that fine-tuning CLIP models degrades their robustness against classifying distribution-shifted samples.

Labeled OOD samples are typically unavailable in real-world applications, and the number of in-distribution samples is usually limited. Therefore, external data have been widely employed (Rafiee et al., 2022) in two ways: a) outlier exposure where the external data is treated as anomalous (Hendrycks et al., 2019a), and b) using models pretrained on auxiliary data (Sun et al., 2022). Outlier exposure leads to performance gains only if the auxiliary data are sufficiently diverse and disjoint from the in-distribution (Hendrycks et al., 2019a; Liznerski et al., 2022). On the other hand, Hendrycks et al. (2020; 2021a) showed that pretrained backbones can enhance OOD detection performance and robustness without relying on dataset-specific shortcuts (Geirhos et al., 2020). Consequently, pretrained models are suitable candidates for OOD detection, while Galil et al. (2023) has recently established CLIP models for OOD detection, especially when in-distribution label names are present (Ming et al., 2022).

In parallel, several OOD detection methods still rely on similar small-scale benchmarks based on low-resolution images (Mohseni et al., 2021; Rafiee et al., 2022; Esmaeilpour et al., 2022), such as CIFAR (Krizhevsky et al., 2009). Huang & Li (2021) argued that methods explicitly tuned for these benchmarks may not always translate effectively into larger-scale and real-life applications. Towards this direction, new large-scale and more challenging benchmarks have been introduced (Hendrycks et al., 2021b; Yang et al., 2022; Bitterwolf et al., 2023), which consider ImageNet (Deng et al., 2009) as in-distribution. Finally, even though the robustness against adversarial attacks has been sufficiently explored in image classification (Szegedy et al., 2014; Goodfellow et al., 2015; Mao et al., 2023), less attention has been given to studying the construction of robust visual OOD detectors (Azizmalayeri et al., 2022; Yin et al., 2021). Even though several advancements in visual feature extractors have been made and new large-scale OOD detection benchmarks (Bitterwolf et al., 2023) have been proposed, limited research has been conducted in OOD detection regarding the choice of pretrained model and evaluation scheme, especially regarding CLIP models (Ming et al., 2022; Galil et al., 2023; Esmaeilpour et al., 2022).

In this paper, we present an experimental study across 25 feature extractors and several visual OOD detection benchmarks. Using the existing publicly available models, we demonstrate that large-scale CLIP models are

robust unsupervised OOD detectors and focus on adapting the representations of CLIP under different OOD detection settings. Under this scope, we examine several OOD detection setups based on the availability of labels or image captions (i.e. in-distribution class names). The core contributions of this work are summarized as follows:

- To investigate whether the dependence between in-distribution accuracy and unsupervised OOD detection performance can be confirmed without in-distribution fine-tuning, we quantify the performance of 25 pretrained models across 4 benchmarks (Fig. 2). CLIP models exhibit the strongest positive correlation without fine-tuning across all benchmarks ($R^2$ coefficient $\geq 0.92$). Interestingly, the features of CLIP ViT-G outperform the ones from supervised ImageNet pretraining on ImageNet-based OOD detection.

- To adapt the representations of CLIP for OOD detection, we propose a simple and scalable method called pseudo-label probing (PLP). The text-based pseudo-labels are computed using its text encoder based on the maximum image-text feature similarity. We leverage the obtained text-based pseudo-labels to train a linear layer on top of CLIP. Without modifying the weights of CLIP or training separate text/image encoders, PLP surpasses the previous state-of-the-art (Ming et al., 2022) on 5 ImageNet benchmarks by an average AUROC gain of 1.8% and 3.4% for CLIP ViT-H and CLIP ViT-G, respectively. Moreover, linear probing achieves superior performance on ImageNet-based OOD benchmarks compared to fine-tuning, notably by a mean AUROC gain of 7.3% using CLIP ViT-H.

- Finally, we show that a simple feature-based adversarial attack tailored for OOD detection can easily fool CLIP ViT-G trained on billions of samples ($86.2\% \rightarrow 50.3\%$ AUROC deterioration) by changes that are invisible to humans.

## 2 Related work

### 2.1 Supervised OOD detection methods

Supervised OOD detection methods rely on the fact that in-distribution classification accuracy is positively correlated with OOD detection performance (Fort et al., 2021; Galil et al., 2023). For that reason, many OOD detection methods derive anomaly scores from supervised in-distribution classifiers. Hendrycks & Gimpel (2017) developed the maximum softmax probability (MSP) as OOD detection score, which is frequently used, or its temperature scaled version (Liang et al., 2018). More recently developed scores based on the logits of the in-distribution classifier are the maximum logit (Hendrycks et al., 2022) or the negative energy scores by Liu et al. (2020). An alternative OOD detection score, which requires the in-distribution labels, is the parametric Mahalanobis-based score (Lee et al., 2018). The Mahalanobis score assumes that the representations from each class are normally distributed around the per-class mean and conform to a mixture of Gaussians (Ren et al., 2021; Fort et al., 2021). Later on, Sun et al. (2022) established an important yet simple OOD detection score, namely the $k$-nearest neighbors (NN) distance, without requiring the feature norms (Tack et al., 2020) or temperature tuning (Rafiee et al., 2022). The $k$-NN distance has the advantage of being non-parametric and model- and distribution-agnostic.

Supervised learning may not always produce sufficiently informative features for identifying OOD samples (Winkens et al., 2020). To this end, additional tasks have been proposed to enrich the supervised-learned features. Examples include determining the key in-distribution transformations and predicting them (Hendrycks et al., 2019b) or contrastive learning. Mohseni et al. (2021) attempt to first learn the domain-specific transformations for each in-distribution using Bayesian optimization. Zhang et al. (2020) present a two-branch framework, where a generative flow-based model and a supervised classifier are jointly trained.

### 2.2 Unsupervised OOD detection methods

Unsupervised OOD detection methods rely on learning in-distribution features, typically accomplished with contrastive or supervised contrastive learning (Sehwag et al., 2021; Khosla et al., 2020). Contrastive-based

methods can be further enhanced by designing hand-crafted transformations that provide an estimate of near OOD data (Rafiee et al., 2022) or by developing better OOD detection scores Tack et al. (2020). For example, Tack et al. (2020) add a transformation prediction objective (i.e. rotation prediction such as $[0°, 90°, 180°, 270°]$). Sehwag et al. (2021) define a simpler contrastive-based OOD detection method, where the Mahalanobis distance is computed in the feature space using the cluster centers of $k$-means (Lloyd, 1982).

### 2.3 OOD detection methods using external data or pretrained models

Early works attempted to use external data to generate examples near the OOD decision boundary or incorporate them for outlier exposure and negative sampling (Hendrycks et al., 2019a; Rafiee et al., 2022). Nonetheless, disjointness between the in and out-distribution cannot be guaranteed, and applying shifting transformations depends on the in-distribution (Mohseni et al., 2021), which limits the applicability of such approaches.

Hendrycks et al. (2020) show that large-scale models pretrained on diverse external datasets can boost OOD detection performance. Recent OOD detection scores, which leverage pretrained models, deal with the large semantic space by, for instance, grouping images with similar concepts into small groups as in Huang & Li (2021). The majority of existing methods focus on supervised OOD detection (Sun et al., 2022; Galil et al., 2023). Such approaches include fine-tuning the whole network or parts of it (Reiss et al., 2021). Contrarily, Ren et al. (2021) showed that a Mahalanobis-based score could achieve comparable OOD detection performance on small-scale benchmarks without fine-tuning.

Out of the limited label-free methods based on pre-trained models, a simple approach is the $k$-NN feature similarity, which has not been studied systematically across pre-trained models (Sun et al., 2022). Another unsupervised approach aims at initially detecting an *a priori* set of visual clusters. The obtained clusters are subsequently used as pseudo-labels for fine-tuning (Cohen et al., 2023). Apart from unsupervised OOD detection, CLIP models can additionally leverage in-distribution class names (Esmaeilpour et al., 2022), and in-distribution prototypes can be obtained using the textual encoder, which has not been thoroughly investigated. Esmaeilpour et al. (2022) extend the CLIP framework by training a text-based generator on top, while Ming et al. (2022) compute the maximum softmax probability of image-text feature similarities. Finally, Wang et al. (2023) hold the current large-scale OOD state-of-the-art using CLIP by training a separate text encoder on 3 million auxiliary text-image pairs. Still, it is unknown whether this approach scales on larger-scale CLIP models. Approaches that work with off-the-shelf-features without requiring external image-text data (Wang et al., 2023), fine-tuning, or training of separate encoders (Wang et al., 2023) remain relatively unexplored, especially in conjunction with CLIP models Wortsman et al. (2022).

### 2.4 OOD detection robustness

Hendrycks & Dietterich (2019) analyzed the robustness under corruptions and geometric perturbations, such as Gaussian noise and brightness shift. Since it is not always clear which manually perturbed images are present in the in-distribution (and which are not), attention has been given to adversarial robustness (Chen et al., 2020a). Existing works have primarily focused on fooling supervised OOD detection methods on small-scale benchmarks (Yin et al., 2021). Even though the zero-shot classification performance of CLIP deteriorates significantly when the input images are constructed adversarially (Mao et al., 2023), their adversarial OOD detection robustness has not been thoroughly investigated. Finally, even though adversarial feature-based methods exist (Guo et al., 2018; Heng et al., 2018; Luo et al., 2022), they have not been applied to OOD detection.

## 3 The proposed OOD detection setup

### 3.1 Considered pretrained models

Several supervised CNNs, such as ResNet50 (He et al., 2016), ConvNext (Liu et al., 2022) and ViT (Dosovitskiy et al., 2021) models trained on ImageNet and ImageNet-21K (Deng et al., 2009; Russakovsky et al., 2015) were utilized. Regarding Imagenet-pretrained self-supervised models, the DINO (Caron et al., 2021), MoCov3

| Architecture | Images/ second | Number of Params (M) |
|---|---|---|
| ResNet50 | 2719 | 24 |
| ConvNext-S | 1576 | 49 |
| ConvNext-B | 1157 | 88 |
| ConvNext-L | 695 | 196 |
| ConvNext-B wide | 888 | 88 |
| ConvNext-L deep | 531 | 200 |
| ConvNext-XXL | 180 | 847 |
| ViT-S/16 | 3210 | 22 |
| ViT-B/16 | 1382 | 86 |
| ViT-L/16 | 475 | 303 |
| ViT-H/14 | 183 | 631 |
| ViT-G/14 | 78 | 1843 |

Table 1: **Number of parameters (in millions) and inference time (images per second on a single GPU) for the utilized network architectures.**

| Dataset | Classes | Train images | Validation images |
|---|---|---|---|
| *Pretraining datasets* | | | |
| ImageNet | 1K | 1.28M | - |
| ImageNet-21K | 21K | 14M | - |
| OpenAI-400M | - | 400M | - |
| LAION-2B | - | 2B | - |
| *In-distribution datasets* | | | |
| CIFAR10 | 10 | 50K | 10K |
| CIFAR100 | 100 | 50K | 10K |
| ImageNet | 1K | 1.28M | 50K |
| *Out-distribution datasets* | | | |
| iNaturalist | 110 | - | 10K |
| SUN | 50 | - | 10K |
| Places | 50 | - | 10K |
| IN-O | 200 | - | 2K |
| NINCO | 64 | - | 5.88K |
| Texture | 47 | - | 5.54K |
| CIFAR10-A | 10 | - | 1000 |
| CIFAR10-AS | 10 | - | 1000 |

Table 2: **An overview of the number of classes and the number of samples on the considered datasets.**

(Chen et al., 2021), and MSN (Assran et al., 2022) were selected. Finally, CLIP-based models were either trained on OpenAI-400M (Radford et al., 2021) or LAION-2B (Schuhmann et al., 2022), which consists of 400M and 2 billion image-text description pairs, respectively. Further information regarding the network complexities is reported in Tab. 1. To quantify the complexity, we performed inference on a single GPU with a batch size of 256 to compute the images per second that were processed at $224 \times 224$ resolution.

## 3.2 Datasets and metrics

We denote the in-distribution as $\mathcal{D}_{\mathrm{in}}$, and the out-distribution as $\mathcal{D}_{\mathrm{out}}$. The corresponding train and test splits are indicated with a superscript. To design a fair comparison between vision and vision-language models, we define *unsupervised OOD detection* without having access to the set of $\mathcal{D}_{\mathrm{in}}$ label names. Following Huang & Li (2021), we use ImageNet as $\mathcal{D}_{\mathrm{in}}$ for large-scale OOD detection benchmarks and CIFAR for small-scale benchmarks.

For the large-scale benchmarks, we use the following 5 OOD datasets: Imagenet-O (IN-O) (Hendrycks et al., 2021b), a flower-based subset of iNaturalist (Van Horn et al., 2018), Texture (Cimpoi et al., 2014), a subset of the SUN scene database (Xiao et al., 2010), a subset of Places (Zhou et al., 2017). IN-O contains 2K samples from ImageNet-21K, excluding ImageNet. It is worth noting that Places, Textures, and IN-O have a class overlap of 59.5%, 25.6%, and 20.5% with ImageNet, according to Bitterwolf et al. (2023). Therefore, we additionally use the newly proposed NINCO Bitterwolf et al. (2023), which contains 5879 samples belonging to 64 classes with zero class overlap with ImageNet's classes. Dataset information is summarized in Tab. 2.

To quantify the OOD detection performance, the area under the receiver operating characteristic curve (AUROC) and the false positive rate at 95% recall (FRP95) are computed between $\mathcal{D}_{\mathrm{out}}^{test}$ test and $\mathcal{D}_{\mathrm{in}}^{test}$. Below, we present the used OOD detection scores, given a pretrained backbone model $g$.

**1-NN.** For the unsupervised evaluations, we use the maximum of the cosine similarity (Sun et al., 2022) between a test image $x'$ and $x_i \in \mathcal{D}_{\text{in}}^{\text{train}} = \{x_1, x_2 \ldots, x_N\}$ as an OOD score:

$$s_{\text{NN}}(x') = \max_i \text{sim}(g(x'), g(x_i)), \tag{1}$$

where $\text{sim}(\cdot)$ is the cosine similarity and $N$ the number of $\mathcal{D}_{\text{in}}^{\text{train}}$ samples .

**Mahalanobis distance (MD).** The MD can be either applied directly on the feature space of the pre-trained model, $z_i = g(x_i)$ or on the trained linear head, $z_i = h(g(x_i))$. However, MD assumes that the in-distribution labels $y_i \in \{y_1, \ldots, y_N\}$ are available. We denote the class index $c \in \{1, \ldots, C\}$, with $C$ being the number of $\mathcal{D}_{\text{in}}$ classes and $N_c$ the number of samples in class $c$. For each class $c$, we fit a Gaussian distribution to the representations $z$ (Lee et al., 2018). Specifically, we first compute the per-class mean $\mu_c = \frac{1}{N_c} \sum_{i:y_i=c} z_i$ and a shared covariance matrix

$$\Sigma = \frac{1}{N} \sum_{c=1}^{C} \sum_{i:y_i=c} (z_i - \mu_c)(z_i - \mu_c)^\top. \tag{2}$$

The Mahalanobis score is then computed for each test sample as

$$\text{MD}_c(z') = \left(z' - \mu_c\right)\Sigma^{-1}\left(z' - \mu_c\right)^\top, \tag{3}$$

$$s_{\text{MD}}(x') = -\min_c \text{MD}_c(z'). \tag{4}$$

MD can also be applied with cluster-wise means, for instance, using the $k$-means cluster centers computed on the feature space of $g$ (Sehwag et al., 2021). We denote this score as $k$-means MD and use the number of ground truth $\mathcal{D}_{\text{in}}$ classes to compute the cluster centers.

**Relative Mahalanobis distance (RMD).** Given the in-distribution mean $\mu_0 = \frac{1}{N} \sum_i^N z_i$, we additionally compute $\Sigma_0 = \frac{1}{N} \sum_i^N (z_i - \mu_0)(z_i - \mu_0)^\top$ to compute $\text{MD}_0$ analogously to Eq. (3). Subsequently, the RMD score Ren et al. (2021) can be defined as

$$s_{\text{RMD}}(x') = -\min_c \left(\text{MD}_c(z') - \text{MD}_0(z')\right). \tag{5}$$

**Energy.** As in Liu et al. (2020), negative free energy is computed over the logits $z$ of an $\mathcal{D}_{\text{in}}$ classifier as

$$s_{\text{energy}}(x') = T \cdot \log \sum_i^C e^{z_i'/T}, \tag{6}$$

where $T = 1$ is a temperature hyperparameter.

### 3.3 Pseudo-Label Probing (PLP) using CLIP's textual encoder

**The pseudo-MSP baseline.** Ming et al. (2022) proposed a simple visual OOD detection method to leverage the text encoder of CLIP by feeding the $\mathcal{D}_{\text{in}}$ class names in parallel with computing the image representations using the image encoder of CLIP. A set of captions such as "a photo of a {*label*}" is used to produce the text representations. The text-based representations are averaged after $L2$ normalization. Subsequently, $L2$ normalized text-image cosine similarities for each test image are computed. The maximum softmax probability (MSP) is used as an OOD score for text-image similarities. This is conceptually similar to using MSP on the logits of a supervised $\mathcal{D}_{\text{in}}$ classifier. We use pseudo-MSP as a baseline.

**PLP.** We propose a simple method called *pseudo-label probing* (PLP) that extends pseudo-MSP. After computing the image-text similarity, we derive a pseudo-label for each $\mathcal{D}_{\text{in}}^{train}$ image using *argmax*, the class index that corresponds to the maximum image-text similarity. To enhance the predictions of CLIP to assign high probabilities to $\mathcal{D}_{\text{in}}^{train}$, a linear layer is then trained with the obtained pseudo-labels on the features of $g$. During the OOD detection evaluation stage, the RMD or energy score is used, namely *PLP + RMD* and *PLP + Energy*.

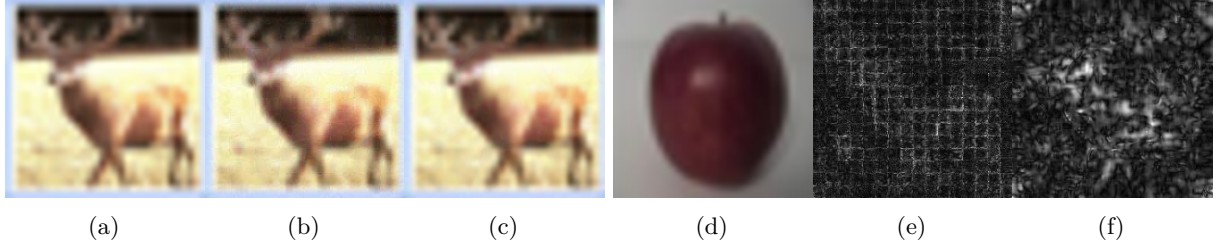

(a)     (b)     (c)     (d)     (e)     (f)

Figure 1: **Generating an adversarial OOD example of a deer that is close enough in the feature space to an in-distribution image of an apple**. From left to right: a) the original OOD image from the CIFAR10 test set, b) the adversarial example without smoothing, c) the adversarial example with smoothing, d) the randomly sampled in-distribution target image from CIFAR100, e) the per-pixel Euclidean distance between the original image and perturbed image, and f) the distance between the original and the smoothly perturbed image.

Similar to pseudo-MSP, PLP remains OOD data-agnostic and assumes no prior information on $\mathcal{D}_{\text{out}}$. PLP is conceptually similar to self-training Xie et al. (2020). Differently from standard self-training, we only train a linear layer using the text-based prototype that is closest to each $\mathcal{D}_{\text{in}}$ image in feature space derived from the textual encoder of CLIP. Finally, while both pseudo-MSP and PLP do not modify the parameters of CLIP or train external image/text encoders Esmaeilpour et al. (2022); Wang et al. (2023), we show that PLP scales better with larger-scale models (Section 4) while training a linear layer adds minimal overhead compared to pseudo-MSP.

We additionally consider supervised linear probing as an alternative to the existing paradigm of fine-tuning visual feature extractors. Linear probing refers to training a linear head on the features of the backbone $g$, using the $\mathcal{D}_{\text{in}}^{\text{train}}$ labels and acts as an upper bound for PLP. Subsequently, RMD or Energy is computed. This is in contrast to existing approaches (Fort et al., 2021; Huang & Li, 2021) that typically fine-tune the visual backbone, which is significantly slower, computationally more expensive, and may lead to catastrophic forgetting (Kemker et al., 2018).

### 3.4 Feature-based adversarial OOD data manipulation

The unsupervised OOD detection performance of CLIP ViT-G, combined with its known classification robustness against natural distribution shifts, raises the question of whether these models are also adversarially robust OOD detectors. To expose the vulnerability of CLIP as an OOD detector, we develop a feature-based adversarial data manipulation tailored for OOD detection similar to Guo et al. (2018); Heng et al. (2018); Luo et al. (2022). We solely aim to test adversarial OOD detection; thus, we do not compare it with feature-based methods that can be modified for OOD detection.

For a test image $x' \in \mathcal{D}_{\text{out}}^{\text{test}}$, we randomly choose an in-distribution image $x \in \mathcal{D}_{\text{in}}^{\text{train}}$ as the target. Instead of maximizing the OOD scores directly (Azizmalayeri et al., 2022; Chen et al., 2020a), we create an adversarial perturbation $\rho$ with the same dimensions as $x'$ that maximizes the cosine similarity between the in-distribution feature $g(x)$ and $g(x' + \rho)$. We use the Adam optimizer to compute $\rho$ by minimizing $-\sin(g(x), g(x' + \rho))$, starting with Gaussian noise $\rho \sim \mathcal{N}(0, 10^{-3})$ and clipping $x' + \rho$ to the pixel range $[0, 1]$ after every update step, similar to Yin et al. (2021). We emphasize that we do not directly restrict the perturbation size and only limit the number of steps, as opposed to Yin et al. (2021).

We experimentally observe that in the case of ViTs, the perturbations are quite visible along the edges of the transformer patches (Fig. 1). To create more natural appearing adversarial examples, we enforce the smoothness of the perturbation by regularizing the allowed perturbation difference between neighboring pixels. We compute the image gradient $\partial\rho/\partial h$ and $\partial\rho/\partial w$ in the horizontal and vertical direction, respectively. The image gradients have the same shape as the image, $3 \times H \times W$, and we define the regularization term as

$$\ell_{\text{smooth}}(\rho) = \frac{1}{3HW} \sum_{ijk} \left(\frac{\partial\rho}{\partial h}\right)^2_{ijk} + \left(\frac{\partial\rho}{\partial w}\right)^2_{ijk}, \tag{7}$$

| Method | OOD Dataset | | | | | | | | | |
| | iNaturalist Plants | | SUN | | IN-O | | Texture | | NINCO | |
| | FPR95↓ | AUROC↑ | FPR95↓ | AUROC↑ | FPR95↓ | AUROC↑ | FPR95↓ | AUROC↑ | FPR95↓ | AUROC↑ |
| **CLIP ViT-B** | | | | | | | | | | |
| Ming et al. (2022) | **30.91** | **94.61** | **37.59** | **92.57** | - | - | 57.77 | 86.11 | - | - |
| Pseudo-MSP† | 52.10 | 89.55 | 40.34 | 91.92 | 78.20 | 79.47 | 55.28 | **86.82** | 83.09 | 72.44 |
| PLP + Energy | 47.70 | 93.36 | 58.15 | 88.69 | **79.00** | **80.72** | 54.63 | 86.44 | **81.93** | **75.65** |
| **CLIP ViT-L** | | | | | | | | | | |
| Ming et al. (2022) | 28.38 | 94.95 | **29.00** | **94.14** | - | - | 59.88 | 84.88 | - | - |
| Pseudo-MSP† | 48.37 | 90.98 | 29.85 | 93.91 | 72.15 | 81.49 | 57.92 | 85.42 | 74.72 | 78.47 |
| PLP + Energy | **27.24** | **95.40** | 44.99 | 91.05 | **64.70** | **85.81** | **46.75** | **88.22** | **71.95** | **83.83** |
| **CLIP ViT-H (LAION-2B)** | | | | | | | | | | |
| Pseudo-MSP† | 60.35 | 90.45 | 42.92 | 91.50 | 59.65 | 87.00 | 46.95 | 89.67 | 74.24 | 82.99 |
| PLP + Energy | **48.61** | **93.02** | **40.90** | **91.78** | **52.50** | **89.24** | **35.00** | **91.96** | **71.9** | **84.88** |
| **CLIP ViT-G (LAION-2B)** | | | | | | | | | | |
| Pseudo-MSP† | 56.49 | 90.55 | 38.69 | 92.75 | 55.50 | 88.11 | 46.95 | 89.70 | 71.30 | 83.38 |
| PLP + Energy | **23.51** | **96.01** | **34.41** | **92.90** | **42.80** | **91.39** | **30.38** | **92.74** | **59.32** | **88.65** |

Table 3: **OOD detection performance metrics using CLIP models and label names for ImageNet-1K as in-distribution**. We report the best metric in bold across different ViTs. The symbol † indicates our reproduction of Ming et al. (2022).

where $i, j, k$ run over image dimensions. We then minimize the loss

$$\ell_{\text{adv}} = -\text{sim}(g(x), g(x' + \rho)) + \lambda \ell_{\text{smooth}}(\rho), \tag{8}$$

with respect to the pertubation $\rho$, where $\lambda$ is a hyperparameter. During the evaluation, we remove the chosen target image, $x$, from $\mathcal{D}_{\text{in}}^{\text{train}}$ to show that the adversarial example, $x' + \rho$, cannot be detected as OOD from the remaining in-distribution examples. As a proof of concept, we create two adversarial OOD datasets[1] for the CIFAR100 → CIFAR10 benchmark, namely CIFAR10-A ($\lambda = 0$) and its smoothened version CIFAR10-AS ($\lambda > 0$). The generation of an adversarial example is shown in Fig. 1. More adversarial examples can be found in the supplementary material.

### 3.5 Experimental evaluations

First, we benchmark 25 publicly available pretrained models on common ImageNet and CIFAR OOD detection benchmarks, as illustrated in Fig. 2. Second, we compare PLP against the aforementioned baselines on ImageNet benchmarks in Tab. 3. For a fair comparison with Ming et al. (2022), we use the CLIP ViT-B and ViT-L trained on OpenAI-400M Radford et al. (2021), but also report results with the recently released CLIP ViT-G trained on LAION-2B (Cherti et al., 2022). In contrast to Huang & Li (2021), we remove the Places OOD subset due to its high class overlap of 59.5% with ImageNet (Bitterwolf et al., 2023); instead, we add IN-O and NINCO. Third, in Tab. 4, we compare linear probing on CLIP's visual representations to standard supervised fine-tuning for OOD detection. Fourth, in Tab. 5, we conduct further OOD detection evaluations with CLIP ViT-G, based on the availability of $\mathcal{D}_{\text{in}}$ class names or (few-shot) labeled images and different detection scores. In Tab. 6, we compare linear probing to fine-tuning for ImageNet-21K pretrained models. Finally, we study the robustness against the adversarially created OOD samples (CIFAR10-A, CIFAR10-AS) using CIFAR100 as $\mathcal{D}_{\text{in}}$.

### 3.6 Implementation details

Since probing and PLP only train a linear layer on precomputed representations, it is more scalable and significantly faster than fine-tuning while having minimal overhead compared to pseudo-MSP. We used the Adam optimizer (Kingma & Ba, 2014) with a mini-batch size of 256 for CIFAR10 and CIFAR100 and 8192 for ImageNet and trained for 100 epochs with a weight decay of $10^{-3}$. The learning rate is set to $10^{-3} \cdot (\text{mini-batch size})/256$ with a linear warm-up over the first ten epochs and cosine decay after that. All

---

[1] https://drive.google.com/drive/folders/1pYGEPQwagRzdKlPqMQv7sRH6C9MV5vNF

| Method | iNaturalist Plants | | SUN | | OOD Dataset
Places | | Texture | | NINCO | |
|---|---|---|---|---|---|---|---|---|---|---|
| | FPR95↓ | AUROC↑ | FPR95↓ | AUROC↑ | FPR95↓ | AUROC↑ | FPR95↓ | AUROC↑ | FPR95↓ | AUROC↑ |
| **CLIP ViT-B** | | | | | | | | | | |
| Finetune + Energy | 42.60 | 83.95 | 60.87 | 73.12 | 64.39 | 69.65 | 61.16 | 73.18 | 67.57 | 70.36 |
| Finetune + MSP | 36.80 | 90.49 | 60.53 | 81.72 | 63.29 | 80.54 | 54.42 | 82.60 | **63.70** | **80.94** |
| Probing + Energy | **24.36** | **95.72** | **50.30** | **90.03** | **50.26** | **88.60** | **50.97** | **88.10** | 78.31 | 79.23 |
| **CLIP ViT-L** | | | | | | | | | | |
| Finetune + Energy | 30.35 | 88.75 | 46.75 | 79.42 | 54.60 | 74.11 | 53.03 | 76.37 | 55.70 | 77.70 |
| Finetune + MSP | 31.33 | 91.90 | 51.33 | 84.97 | 55.68 | 82.90 | 48.68 | 84.52 | **56.79** | 84.27 |
| Probing + Energy | **8.65** | **97.89** | **41.42** | **91.91** | **43.05** | **90.81** | **44.14** | **90.27** | 67.05 | **86.32** |
| **CLIP ViT-H (LAION-2B)** | | | | | | | | | | |
| Finetune + Energy | 27.99 | 90.07 | 39.77 | 83.27 | 49.66 | 77.84 | 42.37 | 81.52 | **51.62** | 78.18 |
| Finetune + MSP | 26.70 | 92.90 | 49.19 | 85.69 | 54.51 | 83.28 | 46.66 | 84.70 | 55.48 | 83.90 |
| Probing + Energy | **6.46** | **98.28** | **32.42** | **93.47** | **38.89** | **91.68** | **26.45** | **93.93** | 55.39 | **89.79** |
| **CLIP ViT-G (LAION-2B)** | | | | | | | | | | |
| Probing + Energy | 6.29 | 98.27 | 32.21 | 93.27 | 37.16 | 92.02 | 24.63 | 94.31 | 49.35 | 90.4 |

Table 4: **Supervised OOD detection performance on ImageNet-1K using CLIP: linear probing outperforms fine-tuning on average, especially for larger-scale models, while being computationally cheaper and significantly faster to train**. The best method for each model architecture is bolded, excluding CLIP ViT-G. The reported results from fine-tuning are based on our evaluations using publicly available CLIP weights fine-tuned on ImageNet-1K from *timm* Wightman (2019).

the experiments were carried out in a single NVIDIA A100 with 40GB VRAM. Moreover, we emphasize that the standard deviation of probing and PLP is less than 0.01%, measured over 10 independent runs. To create the adversarial datasets CIFAR10-A and CIFAR10-AS, we perform 250 steps with the Adam optimizer with a learning rate of $10^{-3}$ on 1K OOD images. We set $\lambda$ to $5 \cdot 10^3$ when applying smoothing (Eq. 8).

## 4 Experimental results

**Unsupervised OOD detection.** In Fig. 2, we initially investigate whether there is a connection between the $\mathcal{D}_{\text{in}}^{test}$ classification accuracy and unsupervised OOD detection AUROC by benchmarking 25 feature extractors. Out of them, CLIP models exhibit the strongest correlation ($R^2$ coefficient $\geq 0.92$) independent of their network architecture (i.e. ConvNext, ViTs, etc). CLIP's best instances (ViT-G, ConvNext-XXL) are currently the best-performing unsupervised OOD detectors, aligning with recent results from Galil et al. (2023). The features of CLIP ViT-G even outperform the ones from supervised training on ImageNet as $\mathcal{D}_{\text{in}}$. Moreover, we observe that when the $\mathcal{D}_{\text{pretrain}}$ is different from $\mathcal{D}_{\text{in}}$ (CIFAR benchmarks), a positive correlation can still be identified for self-supervised and supervised models pretrained on ImageNet ($R^2 = 0.82$), yet not as strong as CLIP ($R^2 = 0.95$).

**Large-scale OOD detection with access to class names.** We report absolute gains and AUROC scores. In Tab. 3, we compare pseudo-MSP, based on our reproduction of Ming et al. (2022), to PLP on all five large-scale benchmarks and find an average improvement of 2.81%, 1.85%, 3.44% AUROC for CLIP ViT-L, ViT-H, and ViT-G, respectively. The reported gains are computed using the energy score, which is slightly superior to RMD on average and significantly faster to compute. Interestingly, PLP has consistent improvements on the newly proposed NINCO dataset, wherein all samples have been visually verified to be semantically different from $\mathcal{D}_{\text{in}}$, with an average AUROC gain of 3.93% across CLIP models. We highlight that the discrepancy between our reproduction and the reported results of Ming et al. (2022) using ViT-B and ViT-L is *not* due to different CLIP weights. We provide a detailed analysis of our reproduction in the supplementary and show that we were not able to get the same results with Ming et al. (2022).

**Large-scale supervised OOD detection.** As shown in Tab. 4, fine-tuning CLIP models is often unnecessary for supervised OOD detection. More precisely, probing on larger models such as CLIP ViT-H consistently outperforms fine-tuning by a large margin of 7.34% on average. The performance discrepancy of CLIP ViT-L

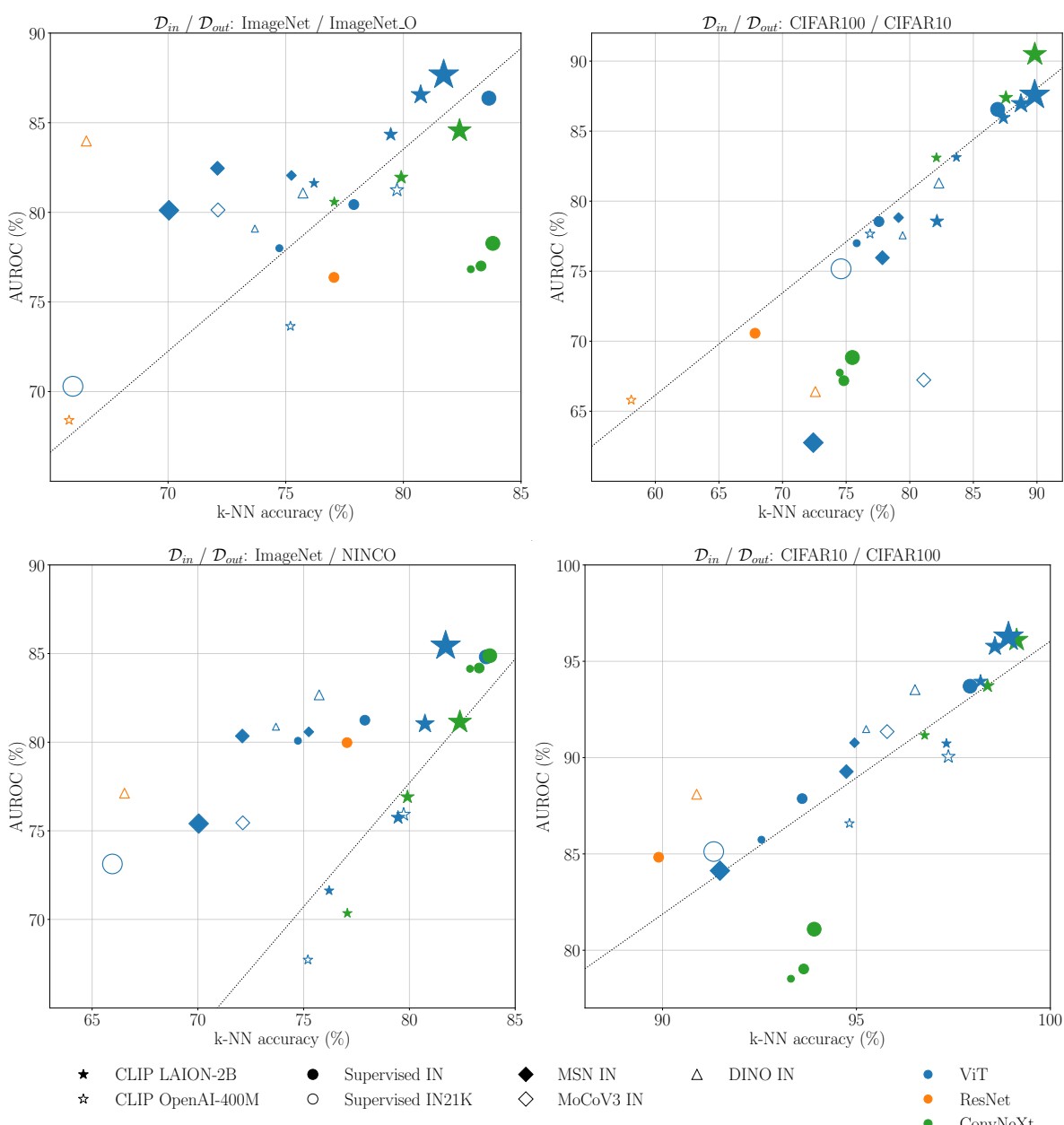

Figure 2: **In-distribution test accuracy using $k$=20 nearest neighbours (k-NN) (x-axis) versus unsupervised out-of-distribution (OOD) detection score (AUROC %) (y-axis) for ImageNet→ImageNet-O, CIFAR100→CIFAR10, ImageNet→NINCO and CIFAR10→CIFAR100** Deng et al. (2009); Hendrycks et al. (2021b); Krizhevsky et al. (2009). CLIP models Radford et al. (2021); Cherti et al. (2022) exhibit a strong dependence between OOD performance and in-distribution test accuracy even without fine-tuning. The black dotted line is fitted using the CLIP models (starred datapoints), where a coefficient $R^2 > 0.92$ for all benchmarks was found. The OOD detection score is computed using the top-1 NN cosine similarity. Different colors are utilized for different architectures (ViT Dosovitskiy et al. (2021), ConvNeXt Liu et al. (2022), ResNet He et al. (2016)) while symbol sizes roughly indicate architecture size (i.e. Small, Base, Large, Huge, Giga). IN indicates ImageNet and IN21K indicates ImageNet-21K Russakovsky et al. (2015). Best viewed in color.

| CLIP ViT-G/14 | $\mathcal{D}_{\text{in}}$ labels/names | $\mathcal{D}_{\text{in}}$:CIFAR100 $\mathcal{D}_{\text{out}}$:CIFAR10 | CIFAR10 CIFAR100 | ImageNet IN-O | ImageNet NINCO |
|---|---|---|---|---|---|
| $k$-means MD | ✗/ ✗ | 72.8 | 89.5 | 87.6 | 80.7 |
| 1-NN | ✗/ ✗ | **87.6** | **98.2** | **88.0** | **84.0** |
| Pseudo-MSP | ✗/ ✓ | 94.2 | 97.3 | 88.1 | 83.4 |
| PLP + MSP | ✗/ ✓ | 92.7 | 97.9 | 86.6 | 86.7 |
| PLP + RMD | ✗/ ✓ | **97.1** | 98.3 | **91.9** | 88.4 |
| PLP + Energy | ✗/ ✓ | 95.0 | **98.5** | 91.4 | **88.7** |
| MD | ✓/ ✓ | 73.1 | 91.1 | 88.1 | 81.5 |
| RMD | ✓/ ✓ | 96.3 | 98.8 | 92.4 | 89.3 |
| Few-shot $p = 10$ + MSP | ✓/ ✓ | 89.4 | 96.5 | 88.1 | 84.8 |
| Few-shot $p = 10$ + Energy | ✓/ ✓ | 90.9 | 96.6 | 90.8 | 83.0 |
| Probing + MSP | ✓/ ✓ | 94.1 | 98.7 | 88.1 | 89.1 |
| Probing + RMD | ✓/ ✓ | **97.3** | 98.8 | 92.5 | 89.5 |
| Probing + Energy | ✓/ ✓ | 96.3 | **99.1** | **92.9** | **90.4** |

Table 5: **OOD detection AUROCs (%) for multiple evaluations and scores using CLIP ViT-G/14 trained on LAION-2B.**

on Places and SUN compared to the other benchmarks is partially attributed to the class overlap. By contrast, iNaturalist, Texture, and NINCO are sufficiently disjoint from ImageNet in terms of $\mathcal{D}_{\text{in}}$ class overlap.

**Adversarial OOD detection robustness.** By evaluating CLIP ViT-G on the introduced CIFAR100→CIFAR10-A benchmark, we found that it is possible to drop the AUROC score from 87.6% →50.3% using 1-NN and 94.2%→49.5% using pseudo-MSP. Introducing the smoothness restriction (Eq. (8)) degrades performance to 55.8% and 51.9% AUROC on CIFAR100→CIFAR10-AS using 1-NN and pseudo-MSP, respectively. Note that an AUROC score of 50% is a random guess's score, meaning that CLIP ViT-G performs slightly better than a random guess.

**Ablation study for CLIP ViT-G for all OOD detection setups.** In Tab. 5, we conduct additional experimental evaluations using CLIP ViT-G for all three OOD detection scenarios (unsupervised, class names are available, supervised). In the unsupervised case, $k$-means + MD yields an inferior AUROC compared to 1-NN, precisely lower by 6.8% on average. By incorporating the $\mathcal{D}_{\text{in}}$ class names using CLIP's text encoder, we find that the PLP consistently outperforms pseudo-MSP with a mean improvement of 3.16% and 2.63% over PLP+RMD and PLP+Energy respectively. In the supervised scenario, we highlight that RMD is a strong baseline, surpassing MD by 10.75% AUROC on average, while our PLP+Energy marginally improves RMD by 0.47%. MSP is constantly the worst choice as an OOD detection score after linear probing.

## 5 Discussion

**Pixel-related features in CLIP's learned representations.** Similar to Mao et al. (2023) for image classification, the OOD detection performance close to a random guess demonstrates that even the top-performing CLIP models trained on billion-scale image-text pairs can be fooled by a visual signal manipulation that is invisible to humans. This finding gives further evidence that besides label-related features, (local) pixel information affects the learned representations and needs to be explored in greater depth (Park et al., 2023; Dravid et al., 2023). Another possible research avenue is how adversarial OOD samples transfer between different feature extractors, which is left for future work.

**Does PLP improve $\mathcal{D}_{\text{in}}^{\text{test}}$ accuracy?** By comparing the $\mathcal{D}_{\text{in}}^{\text{test}}$ accuracy on ImageNet of the trained head with the pseudo-labels versus the zero-shot classification of CLIP ViT-G, we found a minor accuracy improvement of 0.5% on ImageNet. However, accuracies deteriorate for CLIP ViT-B and CLIP ViT-L on average, which suggests that their $\mathcal{D}_{\text{in}}$ text-based pseudo-labels have less true positives. The fact that PLP improves the OOD detection performance (compared to pseudo-MSP) without significantly increasing

|  | Finetuned on $\mathcal{D}_{\text{in}}$ | $\mathcal{D}_{\text{in}}$:IN1K $\mathcal{D}_{\text{out}}$:NINCO | IN1K IN-O | CIFAR100 CIFAR10 | CIFAR10 CIFAR100 |
|---|---|---|---|---|---|
| ConvNext-B | ✗ | **95.4** | **94.9** | **93.0** | **98.0** |
| ConvNext-B | ✓ | 88.8 | 85.8 | 90.8* | 96.9* |
| ViT-L | ✗ | **95.2** | **95.3** | 89.8 | 94.3 |
| ViT-L | ✓ | 91.3 | 92.1 | **97.9*** | **98.5*** |
| ResNet50+ViT-B | ✗ | **95.9** | **95.8** | 89.2 | 96.6 |
| ResNet50+ViT-B | ✓ | 92.9 | 92.1 | **96.2*** | **98.5*** |

Table 6: **Comparing the OOD detection AUROC (%) of ImageNet-21K pretrained models: linear probing versus fine-tuning.** We use publicly available models with and without fine-tuning on ImageNet from *timm* and evaluate them using the energy score. Following Fort et al. (2021), we use the MD on CIFAR fine-tuned models (indicated with an asterisk), which is inefficient at the scale of ImageNet.

$\mathcal{D}_{\text{in}}^{\text{test}}$ accuracy is a new direction left for future work. Another interesting avenue is whether the text-based pseudo-labels from CLIP can be used with other, possibly smaller-scale, visual feature extractors.

**Does linear probing lead to similar OOD detection performance compared to fine-tuning for ImageNet-21K pretrained models?** In Tab. 6, we compare linear probing versus fine-tuning for 3 publicly available models, where there is a corresponding fine-tuned model on ImageNet. Even though fine-tuning is the standard practice in supervised OOD detection Fort et al. (2021), the obtained results indicate that it performs inferior to linear probing on the ImageNet OOD-related benchmarks even for ImageNet-21K pretrained models. Fine-tuning is, however, still the best approach on the small-scale OOD benchmarks (e.g. CIFAR10 → CIFAR100), given ImageNet21K pretraining. This explains why a simple and scalable approach such as probing may have been overlooked. As suggested by Huang & Li (2021), we confirm that OOD detection approaches tested on small-scale benchmarks do not always translate effectively into larger-scale setups, where simpler methods need to be revisited. Combined with the results from Tab. 4, we validate that the OOD-related information is readily available on large-scale foundational models, as recently stated in Inkawhich et al. (2023); Oquab et al. (2023).

**Is the choice of the best feature extractor consistent for smaller-sized models?** Not in the scale of ViT-B and CIFAR benchmarks. In Fig. 3, we keep the model architecture fixed (ViT-B) and visualize the unsupervised OOD detection performance (1-NN) for different datasets (ImageNet, ImageNet21K, OpenAI-400M, LAION-2B) and pretraining types (CLIP, supervised, self-supervised). The rationale behind using the CIFAR-based benchmarks is to focus on feature transferability. We highlight that large pretraining datasets such as LAION-2B are bottlenecked by the model size of ViT-B as explained in Cherti et al. (2022). Unlike Fig. 2, we show that the ranking of backbones is not always consistent between different $\mathcal{D}_{\text{in}}$ and $\mathcal{D}_{\text{out}}$ at this scale. Finally, DINO ViT-B surpasses supervised ImageNet pretraining on both benchmarks, outlining the transferability of features of self-supervised methods, which is consistent with the results of Ericsson et al. (2021); Naseer et al. (2021); Adaloglou et al. (2023).

## 6 Conclusion

This work presented a thorough experimental study by leveraging pretrained models for visual OOD detection, focusing on CLIP. It was demonstrated that CLIP models are powerful unsupervised OOD detectors, outperforming even $\mathcal{D}_{\text{in}}$ supervised models on large-scale OOD detection benchmarks. A fast, simple, OOD data-agnostic, and scalable method called PLP that trains a linear layer based on $\mathcal{D}_{\text{in}}$ text-based pseudo-labels was introduced. PLP outperformed the previous state-of-the-art (pseudo-MSP) for most CLIP models and ImageNet-based benchmarks while substantially improving the larger architectures (i.e. ViT-G achieves a mean AUROC gain of 3.4%). Furthermore, it was demonstrated that probing can replace the costly fine-tuning on CLIP and even ImageNet-21K models on large-scale benchmarks, where CLIP ViT-H exhibited an average AUROC improvement of 7.3%. Finally, a simple feature-based adversarial OOD data manipulation method showed that billion-scale feature extractors (CLIP ViT-G) are vulnerable to adversarial OOD attacks.

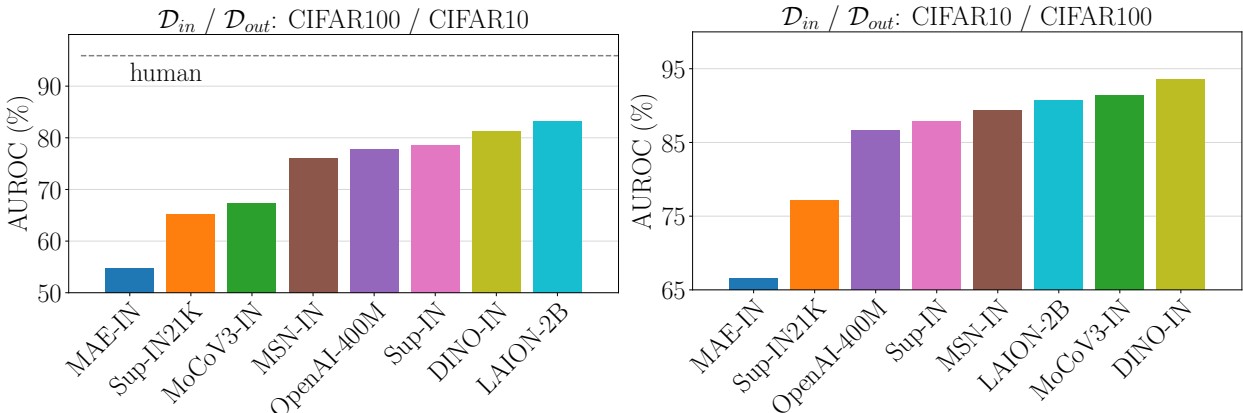

Figure 3: **AUROC values for unsupervised OOD detection using ViT-B/16 pretrained on different datasets (IN, IN-21K, OpenAI-400M, LAION-2B) and pretext tasks**. IN indicates ImageNet. The performance of CLIP on LAION-2B is bottlenecked by the model size as reported in Cherti et al. (2022) and larger networks are needed when scaling up to billion-scale datasets. The horizontal line indicates human-level performance, as reported in Fort et al. (2021).

## 7 Acknowledgements

We would like to thank Georgios Zoumpourlis and Kaspar Senft for their insightful remarks and discussions on the manuscript.

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
