# A Supplementary material: Adapting Contrastive Language-Image Pretrained (CLIP) Models for Out-of-Distribution Detection

## A.1 Partial fine-tuning

To further investigate the behavior of Imagenet-21K pre-trained models presented in Table 6 (main text), we show additional results using partial fine-tuning for the pre-trained ConvNext-B model on CIFAR10 and CIFAR100 in Fig. 1.

## A.2 Additional implementation details

Following Ming et al. (2022), for the text encoder of CLIP, we take the mean of the L2-normalized text representations of the following prompts: "an image of a {label}", "a photo of a {label}", "a blurry photo of a {label}", "a photo of many {label}", "a photo of the large {label}", "a photo of the small {label}". We randomly select $p = 10$ images per class for few-shot probing and take the average AUROC over 5 runs. We set the mini-batch size to 32 for CIFAR100 and 1024 for ImageNet. To enforce reproducibility for our results on Table 6, the corresponding models can be found using the *timm* (version 0.6.12) names Wightman (2019): *vit_large_patch16_224_in21k*, *convnext_base_in22k*, and *vit_base_r50_s16_224_in21k*. We note that probing takes less than 15 minutes on ImageNet on a single GPU.

## A.3 Pseudo-MSP reproducibility issue compared to Ming et al.

Since our reproduction of Pseudo-MSP performs worse than the values reported in Ming et al. (2022), we investigate various combinations of hyperparameters for the method Pseudo-MSP. We compare the CLIP models provided by OpenAI[1] and Hugging Face[2] and find no notable difference. For the label embeddings, we use either the prompt "a photo of a <label>." or the ensemble of five prompts by Ming et al. (2022) as detailed above. In the case of the prompt ensemble, we normalize the text embeddings either before, after, or before and after averaging them. Furthermore, we explore different softmax temperatures in $\{0.01, 0.1, 1, 10\}$, but since every temperature resulted in an improvement of the AUROC score of at most 0.4% compared to the temperature of 1.0, we only report the values for this temperature. Our results are summarised in Table 1 and Table 2. None of the combinations we tried was able to match the performance that Ming et al. reported in their paper.

## A.4 Comparision with CLIPN

In Table 3 we show that PLP does not outperform the recent CLIPN method (Wang et al., 2023). However, we highlight that the recently proposed CLIPN Wang et al. (2023) requires 3M additional image-text-pairs to train a seperate model and it is thus not a fair comparison to Pseudo-MSP and PLP. We include these results to faciliate future comparisons with CLIP ViT-B trained on LAION-2B. It is still unclear if CLIPN can be scaled to larger scale CLIP models as the authors only report results using ViT-B.

## A.5 Does PLP using an MLP achieve superior results?

We found negligible differences when substituting the linear layer with an MLP, which suggests that a linear mapping to the space of in-distribution classes is sufficient for OOD detection.

## A.6 Computational complexity of OOD detection methods.

As reported in Table 6, Fort et al. (2021) used the MD as an OOD score, which is inefficient at the scale of ImageNet and can become prohibitively slow for even larger datasets. Given a feature dimension $d$ and dataset size $N$, the total time complexity scales linearly with dataset size (for the computation of the covariance matrix) and in cubic time with $d$ due to the inverse calculation of the covariance matrix, resulting

---

[1] https://github.com/openai/CLIP

[2] https://huggingface.co/openai/clip-vit-base-patch16 and https://huggingface.co/openai/clip-vit-base-patch16

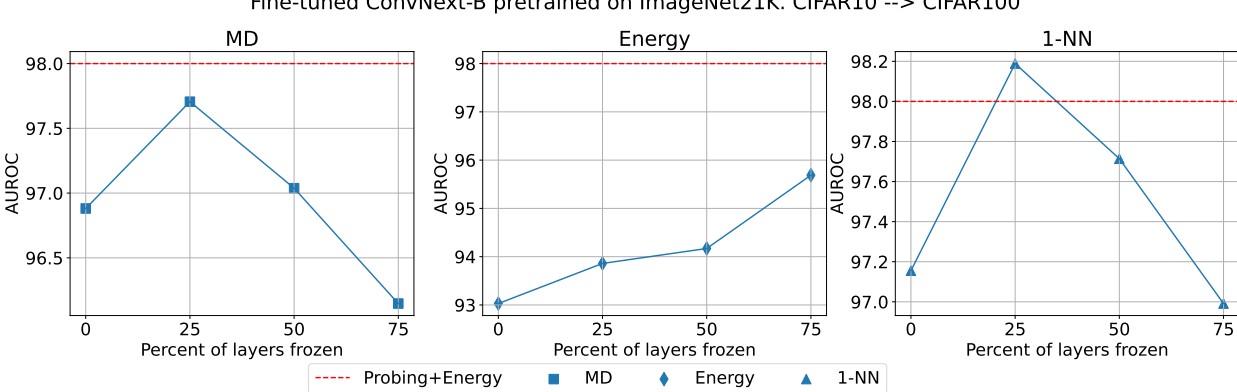

(a) AUROC after CIFAR10 (partial) fine-tuning using ConveNext-B pretrained on ImageNet-21K.

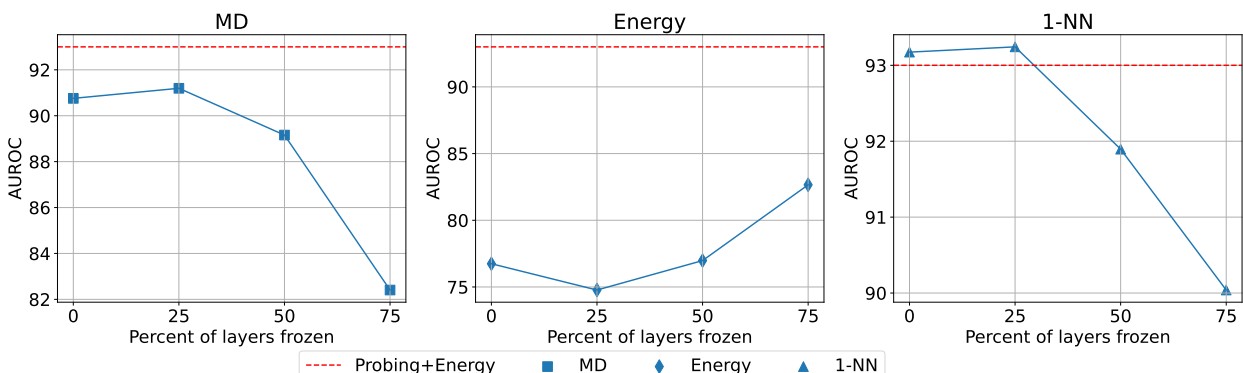

(b) AUROC after CIFAR100 (partial) fine-tuning using ConveNext-B pretrained on ImageNet-21K.

Figure 1: **Benchmarking ImageNet-21K finetuning on CIFAR10→ CIFAR100 (a) and (b) CIFAR100 → CIFAR10 using ConveNext-B weight from timm.**

| OOD Dataset | Prompt | L2 Normalization before/after mean | Weights | AUROC↑ | FPR95↓ |
|---|---|---|---|---|---|
| **Results using CLIP ViT-B/16** | | | | | |
| SUN | ensemble | ✓/✓ | Hugging Face | 91.82 | 40.63 |
| SUN | ensemble | ✓/✓ | OpenAI | 91.92 | 40.34 |
| SUN | ensemble | ✓/✗ | Hugging Face | 91.84 | 40.95 |
| SUN | ensemble | ✓/✗ | OpenAI | 91.94 | 40.36 |
| SUN | ensemble | ✗/✓ | Hugging Face | 91.86 | 40.63 |
| SUN | ensemble | ✗/✓ | OpenAI | 91.96 | **40.27** |
| SUN | single | ✓ | Hugging Face | 91.86 | 42.58 |
| SUN | single | ✓ | OpenAI | **92.00** | 41.71 |
| Texture | ensemble | ✓/✓ | Hugging Face | 86.71 | 55.91 |
| Texture | ensemble | ✓/✓ | OpenAI | 86.82 | 55.28 |
| Texture | ensemble | ✓/✗ | Hugging Face | 86.43 | 56.67 |
| Texture | ensemble | ✓/✗ | OpenAI | 86.55 | 56.15 |
| Texture | ensemble | ✗/✓ | Hugging Face | 86.68 | 56.11 |
| Texture | ensemble | ✗/✓ | OpenAI | **86.79** | **55.48** |
| Texture | single | ✓ | Hugging Face | 85.78 | 60.62 |
| Texture | single | ✓ | OpenAI | 85.93 | 59.97 |
| iNaturalist | ensemble | ✓/✓ | Hugging Face | 89.52 | 51.91 |
| iNaturalist | ensemble | ✓/✓ | OpenAI | 89.55 | 52.10 |
| iNaturalist | ensemble | ✓/✗ | Hugging Face | 90.73 | 47.28 |
| iNaturalist | ensemble | ✓/✗ | OpenAI | 90.75 | 47.54 |
| iNaturalist | ensemble | ✗/✓ | Hugging Face | 89.52 | 51.97 |
| iNaturalist | ensemble | ✗/✓ | OpenAI | 89.55 | 52.17 |
| iNaturalist | single | ✓ | Hugging Face | **92.49** | **41.75** |
| iNaturalist | single | ✓ | OpenAI | **92.49** | 41.96 |

Table 1: Ablation study for different CLIP ViT-B weights on ImageNet OOD detection benchmarks. Our results show that we are not able to reproduce the results from Ming et al. (2022) and that no single method consistently outperforms the others in terms of the prompt to be used and the optimal way to aggregate text prompts.

| OOD Dataset | Prompt | L2 Normalization before/after mean | Implementation | AUROC↑ | FPR95↓ |
|---|---|---|---|---|---|
| **Results using CLIP ViT-L/14** | | | | | |
| SUN | ensemble | ✓/✓ | Hugging Face | 93.91 | 29.78 |
| SUN | ensemble | ✓/✓ | OpenAI | 93.91 | 29.85 |
| SUN | ensemble | ✓/✗ | Hugging Face | 93.87 | 30.72 |
| SUN | ensemble | ✓/✗ | OpenAI | 93.87 | 30.66 |
| SUN | ensemble | ✗/✓ | Hugging Face | **93.93** | **29.62** |
| SUN | ensemble | ✗/✓ | OpenAI | **93.93** | 29.75 |
| SUN | single | ✓ | Hugging Face | 93.25 | 33.86 |
| SUN | single | ✓ | OpenAI | 93.26 | 34.04 |
| Texture | ensemble | ✓/✓ | Hugging Face | 85.41 | 58.51 |
| Texture | ensemble | ✓/✓ | OpenAI | 85.42 | 57.92 |
| Texture | ensemble | ✓/✗ | Hugging Face | 85.44 | 58.38 |
| Texture | ensemble | ✓/✗ | OpenAI | **85.45** | **57.81** |
| Texture | ensemble | ✗/✓ | Hugging Face | 85.42 | 58.49 |
| Texture | ensemble | ✗/✓ | OpenAI | 85.43 | 57.93 |
| Texture | single | ✓ | Hugging Face | 84.48 | 61.00 |
| Texture | single | ✓ | OpenAI | 84.51 | 60.89 |
| iNaturalist | ensemble | ✓/✓ | Hugging Face | 90.99 | 48.58 |
| iNaturalist | ensemble | ✓/✓ | OpenAI | 90.98 | 48.37 |
| iNaturalist | ensemble | ✓/✗ | Hugging Face | 91.27 | 46.98 |
| iNaturalist | ensemble | ✓/✗ | OpenAI | 91.27 | 46.74 |
| iNaturalist | ensemble | ✗/✓ | Hugging Face | 91.00 | 48.34 |
| iNaturalist | ensemble | ✗/✓ | OpenAI | 90.99 | 48.26 |
| iNaturalist | single | ✓ | Hugging Face | 91.95 | 42.89 |
| iNaturalist | single | ✓ | OpenAI | **91.96** | **43.01** |

Table 2: Additional ablation study for different CLIP ViT-L models on ImageNet OOD detection benchmarks.

| Method | OOD Dataset | | | | | | | |
|---|---|---|---|---|---|---|---|---|
| | iNaturalist Plants | | Texture | | SUN | | Places | |
| | FPR95↓ | AUROC↑ | FPR95↓ | AUROC↑ | FPR95↓ | AUROC↑ | FPR95↓ | AUROC↑ |
| Pseudo-MSP | 74.56 | 86.88 | 58.04 | 86.67 | 56.47 | 89.10 | 61.88 | 86.05 |
| PLP + Energy | 61.41 | 92.05 | 45.65 | 88.94 | 57.31 | 88.55 | 59.38 | 85.99 |
| **Methods that use auxialiary image-text data and a train seperate text encoder** | | | | | | | | |
| CLIPN-A | 23.94 | 95.27 | 40.83 | 90.93 | 26.17 | 93.93 | 33.45 | 92.28 |

Table 3: Additional results for ImageNet-based OOD detection benchmarks using CLIP ViT-B/16 trained on LAION-2B. The reported results for CLIPN-A are taken from Wang et al. (2023) while pseudo-MSP is our reproduction of Ming et al. (2022).

in $O(Nd\min(N,d) + d^3) = O(Nd^2 + d^3)$ since $N > d$. We thus conclude that, in addition to the performance metrics, the computational complexities need to be taken into account in future studies to design scalable and efficient OOD detection systems.

### A.7 Additional few-shot evaluations on CIFAR100→ CIFAR10

In Fig. 2, we show that even with 1% of the data, we can surpass the zero-shot 1-NN score. RMD applied on the logits seems to be consistently better than MSP.

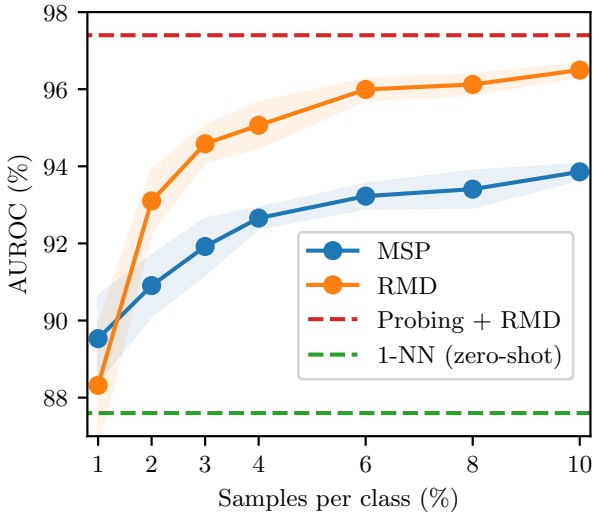

Figure 2: **Few-shot linear probing on CIFAR100 → CIFAR10.** Samples per class (shown as a percentage %) versus OOD detection performance (*y-axis*).

### A.8 Additional adversarial examples

We illustrate more adversarially generated samples using the proposed method in Fig. 3. The created adversarial OOD datasets CIFAR10-A and CIFAR10-AS are publicly available via this hyperlink.

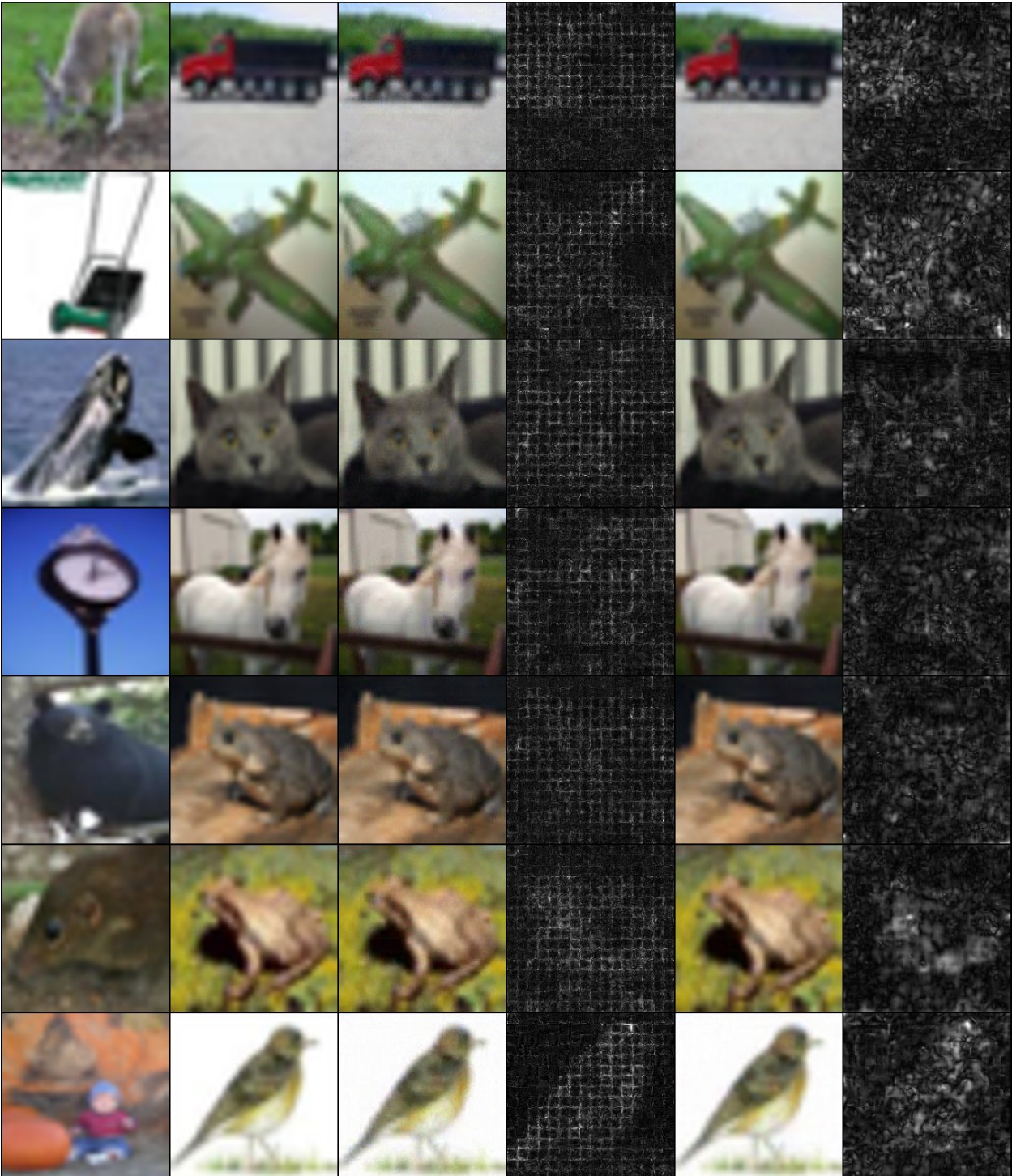

Figure 3: **Additional CIFAR10 adversarially manipulated examples for CIFAR100 → CIFAR10 OOD detection with (CIFAR10-AS) and without (CIFAR10-A) the smoothing constraint.** Columns from *left to right*: target in-distribution image from CIFAR100, original CIFAR10 sample, adversarially manipulated image without smoothing, the Euclidean pixel-wise distance between the original image and perturbed image, adversarial example with smoothing, Euclidean distance.