# OpenReview forum: "Adapting Contrastive Language-Image Pretrained (CLIP) Models for Out-of-Distribution Detection"
_TMLR — Accepted by TMLR_

### Review · Reviewer_FVMe · 2023-11-24

**Summary Of Contributions:**

This paper propose to fine-tune a linear probing layer with the pseudo-label obtained by CLIP to train a visual OOD detector using two existing scores, the RMD and Energy scores. The paper also studies the adversarial image curation by optimizing towards higher cosine similarity discrepancy while keeping the perturbation invisible by adding a regularization term.

**Audience:**

Yes

**Broader Impact Concerns:**

NA.

**Claims And Evidence:**

No

**Requested Changes:**

1. The author says in Section 3.3 that the baselines (Fort et al., 2021; Huang & Li, 2021) differs from the proposed PLP method by fine-tuning the whole visual backbone instead of linear probing. The other components, e.g., the objectives and the pseudo-label, all seem to have been adopted in the baselines. Could the author clarify what is the key contribution of PLP part, as parameter-efficient fine-tuning is a well-known and widely adopted method when scaling method up.

2. Could the author conduct experiments using other parameter-efficient fine-tuning methods like adapter tuning and compare the trade-off between the number of trainable parameters and the performance?

3. Table 3 reports a very low baseline score than its original reported ones in Ming et al. (2022), which the author claims to be caused by the different source weights of the base model. The gap is too large to make me believe it is caused by that the authors downloaded the pretrained weights from a different place. Could the author supplement one experiment by using the same HuggingFace weights as in Ming et al. and compare again?

4. Figure 1 takes up the whole page 2, but it is first mentioned in page 9, which seems to be an unreasonable design, making me very confused when reading through the paper.

5. Why Table 4 compares the fine-tuning with MSP with the proposed Probing with the MSP objective? From Table 5, the Energy objectively is obviously better than MSP. Isn't the fair comparison being Finetune+MSP vs Probing+MSP and Finetune+Energy vs Probing Energy? Please supplement corresponding experiments.

**Strengths And Weaknesses:**

**Strengths**

The paper is well-written and easy to follow. Related works and the experimental settings are clearly presented. The proposed way to curate adversarial examples is also interesting.

**Weakness**

The comparison with the baseline seems unfair and need justification. The author should discuss their contribution compared with the existing works more clearly. See requested changes.

---

> ### Author Response · Authors · 2024-02-14
> **Reply to Reviewer FVMe**
>
> Thank you for the really insightful suggestions and feedback. Below, we aim to address the reviewer’s concerns and remarks.
>
> ## Requested Changes 1: clarify the contribution of PLP
> Pseudo-label probing (PLP) is not parameter-efficient fine-tuning, as the reviewer suggests. The overall idea of the paper is to explore the limits of fast and simple solutions, like training a linear layer on top of the backbone with pseudo-labels derived from the same model. To the best of our knowledge, the idea of CLIP-based pseudo-labels for linear probing has not been used before in the context of OOD detection. Ming et al. only use the maximum softmax probability (MSP) of text-image similarities as the OOD score, while we clearly show that our method (PLP) scales better with larger-scale models while keeping the backbone frozen.
>
>
> Regarding supervised OOD detection, our findings are relatively new to the subfield of OOD detection, where we thoroughly discuss that the common approach is to fine-tune all the layers of the supervised ImageNet or ImageNet-21K pre-trained backbone (Fort et al., 2021; Huang & Li, 2021). By keeping the backbone frozen, we are able to develop an OOD detector of similar or superior performance.
>
> ## Requested Changes 2:Trade-off between parameters and the OOD performance
> To further investigate the presented results in Table 6, we added experiments using partial fine-tuning in the supplementary (Figure 1) on CIFAR10 and CIFAR100 using ConvNext-B pretrained on ImageNet-21K. We show that linear probing+energy is the best-performing method in the vast majority of cases, independent of the OOD score (MD, Energy, 1-NN) for the (partially) fine-tuned models.
>
> Image link: https://ibb.co/DL2X1wt (included in the supplementary)
>
> We hope that the added figures in the supplementary address the reviewer’s concern. Since computing is not a big bottleneck when fine-tuning on CIFAR10 and CIFAR100, we leave parameter-efficient fine-tuning on ImageNet for future work.
>
> ## Requested Changes 3:Reproducibility issue compared to Ming et al.
> We were not able to reproduce the results from Ming et al.. Even after contacting the authors for further clarifications, we were not able to reproduce their reported performance with all different publicly available CLIP weights. Their GitHub repository (https://github.com/deeplearning-wisc/MCM) shows that other researchers have problems reproducing their results. More precisely, their public repository does not support multiple prompt averaging. We provide a more detailed analysis of this in the supplementary. We kindly ask the author to take a look at the added Supplementary Tables. We provide a short version of the tables using CLIP ViT-B below on the ImageNet-iNaturalist OOD benchmark. We will reflect the above in the manuscript, and we thank the reviewer for pointing this out.
>
> | OOD Dataset | Prompt | L2 Normalization before/after mean | Weights | AUROC ↑ | FPR95 ↓ |
> |-|-|-|-|-|-|
> | **Results using CLIP ViT-B/16** | | | | | |
> | iNaturalist | ensemble | ✔️/✔️ | Hugging Face | 89.52 | 51.91 |
> | iNaturalist | ensemble | ✔️/✔️ | OpenAI | 89.55 | 52.10 |
> | iNaturalist | ensemble | ✔️/❌ | Hugging Face | 90.73 | 47.28 |
> | iNaturalist | ensemble | ✔️/❌ | OpenAI | 90.75 | 47.54 |
> | iNaturalist | ensemble | ❌/✔️ | Hugging Face | 89.52 | 51.97 |
> | iNaturalist | ensemble | ❌/✔️ | OpenAI | 89.55 | 52.17 |
> | iNaturalist | single | ✔️ | Hugging Face | **92.49** | **41.75** |
> | iNaturalist | single | ✔️ | OpenAI | **92.49** | 41.96 |
>
> ## Requested changes number 4: We moved Figure 1 to the experimental section.
> ## Requested changes number 5:
> We only included Finetune+MSP because it was the best-performing OOD method when using the fine-tuned CLIP models from the timm library. Below, we include the Finetune+Energy results (AUROC scores), and we will also add them in the supplementary while clarifying that we choose the best out of MSP and Energy score for the considered publicly available CLIP models fine-tuned on Imagenet. Regarding supervised linear probing, our results in Table 5 clearly illustrate that Energy is the best score.
>
> #### AUROC values only are shown below
> | Method | iNat|SUN | Places | Texture  | NINCO |
> |-|----|-------|---|--|---|
> | **CLIP ViT-B**  	|     	| | | 	| 	|
> | Finetune + Energy	| 83.95      	| 73.12    	| 69.65  | 73.18        	| 70.36       	|
> | Finetune + MSP   	| 90.49                    	| 81.72	| 80.54       	| 82.60        	| **80.94**   	|
> | Probing + Energy 	| **95.72**  	| **90.03**	| **88.60**   	| **88.10**    	| 79.23       	|
> | **CLIP ViT-L**  	|    |  |     	|              	|             	|
> | Finetune + Energy	| 88.75                    	| 79.42    	| 74.11       	| 76.37        	| 77.70       	|
> | Finetune + MSP   	| 91.90                    	| 84.97    	| 82.90       	| 84.52        	| 84.27   	|
> | Probing + Energy 	| **97.89**                	| **91.91**	| **90.81**   	| **90.27**    	| **86.32**   	|

---

### Review · Reviewer_V5a6 · 2023-12-21

**Summary Of Contributions:**

The paper presents a comprehensive experimental study on pre-trained feature extractors such as CLIP models for visual out-of-distribution (OOD) detection. Without fine-tuning on the training data, the authors establish a strong positive correlation (R2 ≥ 0.92) between in-distribution classification and unsupervised OOD detection for CLIP models across four benchmarks. They introduce a new and scalable method called pseudo-label probing (PLP), which utilizes label names of the training set to train a linear layer using pseudo-labels derived from the text encoder of CLIP for OOD detection. The authors develop a novel feature-based adversarial OOD data manipulation approach to evaluate the OOD detection robustness of pre-trained models. The study shows that PLP surpasses the previous best method on all five large-scale ImageNet benchmarks, achieving a 3.4% AUROC gain with ViT-G. Linear probing outperforms fine-tuning for CLIP architectures, with ViT-H gaining 7.3% AUROC on average across all ImageNet benchmarks. However, billion-parameter CLIP models struggle to detect adversarially manipulated out-of-distribution images.

**Audience:**

Yes

**Claims And Evidence:**

Yes

**Requested Changes:**

Please see the weaknesses above.

**Strengths And Weaknesses:**

Strengths:
------------

1. As large-scale pre-trained models/foundation models are getting widely adopted it is important to study if they can also be effective in unsupervised OOD detection. The study establishes that large-scale CLIP models serve as robust unsupervised OOD detectors without the need for in-distribution fine-tuning. The strong positive correlation (R2 coefficient ≥ 0.92) between in-distribution accuracy and unsupervised OOD detection across multiple benchmarks underscores the effectiveness of CLIP models.


2. The paper also introduces a novel method, pseudo-label probing (PLP),  a simple and scalable method to adapt CLIP representations for OOD detection. Leveraging text-based pseudo-labels computed using the text encoder, PLP outperforms the previous state-of-the-art method on ImageNet benchmarks, showcasing its effectiveness in enhancing OOD detection performance.

3. The paper presents a novel method for adversarially manipulating out-of-distribution images, revealing that CLIP ViT-G, trained on billions of samples, shows substantial vulnerability to these perturbations with an AUROC deterioration from 86.2% to 50.3% due to imperceptible changes.


Weaknesses
----------------
1. The paper lacks a deeper investigation of the observed results. In particular, it would be helpful to have further experiments to explain why should one expect CLIP embeddings to work well for OOD detection. Evaluation is all on natural images ( that are plentiful on the internet) and CLIP has likely been trained on these or similar images. Is it the wide range of training data that makes them suitable for OOD detection or is it the architecture or training procedure? Would similar claims be valid for CLIP models trained for specific domains e.g. https://github.com/Mauville/MedCLIP?

2. I found the discussion of PLP (section 3.3.) to be inadequate and it is written with the assumption that the reader is familiar with the prior work (Ming et al. 2022). It would be helpful to provide the necessary background and the details of the method and also discuss the differences with Ming et al. 2022.

---

> ### Author Response · Authors · 2024-02-13
> **Reply to reviewer V5a6**
>
> Thank you for the insightful suggestions and feedback. Below, we aim to address the reviewer’s concerns and remarks.
>
> ## Weakness 1A: The paper lacks a deeper investigation of the observed results.
> We agree with the reviewer that a more in-depth investigation of why CLIP embeddings work well for OOD detection is needed. We are the first to provide experimental evidence that using only the image-only embeddings of CLIP is enough to outperform supervised ImageNet-pre-trained models on unsupervised large-scale OOD detection benchmarks, where imagenet is the in-distribution. Prior works mainly investigate the zero-shot classification performance of CLIP, which requires the label names and gives CLIP methods an advantage over image-only supervised and self-supervised feature extractors. We believe this will encourage future works to investigate the critical components and factors that led to the observed superiority in performance.
>
> We can’t answer whether the wide range of training data or the training procedure makes CLIP models suitable for OOD detection based on our experimental study. Our study clearly shows how CLIP can be applied to the 3 different OOD setups: unsupervised, using class names, and supervised OOD detection. To explain the factors that contribute to the superiority of CLIP in OOD detection, one would need to train CLIP with different versions of the data or try different variants of contrastive learning for training. This is a very active field of study that is investigated from concurrent works [3,4], with a focus on Imagenet zero-shot classification accuracy, but requires multiple large-scale training iterations, which is out of the scope of this work.
>
>
>
> ## Weakness 1B: applicability to the medical domain
>
> In principle, our method does not require any modification apart from domain-related prompts.  The main reason new methods do not test different domains, such as the medical domain, is that there are very few OOD benchmarks, and state-of-the-art methods are developed on natural images. PLP only requires a set of prompts; the same ones a practitioner would use for zero-shot classification can be used for OOD detection. Other works based on CLIP designed hand-crafter prompts that are specific to the downstream dataset. We believe that the simplicity of our method, along with the fact that we apply zero prompt engineering, makes our method very attractive to new domains, unlike other works that train text domain-specific text encoders using additional labeled image-text pairs [1,2].
>
>
> ## Weakness 2:
> We will address the remark by revising Sec 3.3 to be more self-complete and clarify our key differences compared to Ming et al. We kindly ask the reviewer to confirm this in the revised manuscript.
>
> ## References
> [1] Esmaeilpour, S., Liu, B., Robertson, E., & Shu, L. (2022, June). Zero-shot out-of-distribution detection based on the pre-trained model clip. In Proceedings of the AAAI conference on artificial intelligence (Vol. 36, No. 6, pp. 6568-6576).
>
> [2] Wang, H., Li, Y., Yao, H., & Li, X. (2023). Clipn for zero-shot ood detection: Teaching clip to say no. In Proceedings of the IEEE/CVF International Conference on Computer Vision (pp. 1802-1812).
>
>
> [3] Mayilvahanan, P., Wiedemer, T., Rusak, E., Bethge, M., & Brendel, W. (2023). Does CLIP's Generalization Performance Mainly Stem from High Train-Test Similarity?. arXiv preprint arXiv:2310.09562.
>
> [4] Xu, H., Xie, S., Tan, X. E., Huang, P. Y., Howes, R., Sharma, V., ... & Feichtenhofer, C. (2023). Demystifying clip data. arXiv preprint arXiv:2309.16671.

---

### Review · Reviewer_a6kb · 2024-01-21

**Summary Of Contributions:**

The authors note that CLIP's performance on ImageNet is correlated with its OOD detection performance. Based on this finding, they propose using CLIP features for OOD detection. In that sense they find linear probing to be superior than fine-tuning. They focus on unsupervised OOD detection, since labels are not available, they follow the procedure introduced by Ming et al. (2022), where ImageNet label names are compared to in-distribution images in order to derive pseudo-labels. The authors explore whether the capabilities of CLIP extend to adversarial detection, concluding this is not the case. The authors include ablations and discussion including different pre-training tasks and datasets.

**Audience:**

Yes

**Claims And Evidence:**

No

**Requested Changes:**

* The authors claim to "introduce a novel method that adversarially manipulates OOD images by matching their representations" without a proper quantitative evaluation, ablation and comparison with other adversarial methods. For example, why not using any of these already existing smoothing techniques? [B,C,D]. Another option would be that instead of "introducing a novel method", you  "show OOD detection with CLIP is vulnerable to adversarial attacks". (until this point is resolved I am going to check "no" at whether the "the claims made in the submission supported by accurate, convincing and clear evidence?")
* Could you update the text to reflect how your work compares to [A]? Note I am not asking for "beating" [A] but at least putting your submission in context of more recent work and adding some discussion about how your works compare and what are the advantages of your method (e.g. not needing to learn a "no" text encoder).
* The linear relation between ImageNet and OOD detection performance is very similar to the effective robustness curve shown in the original CLIP paper (e.g. Figure 13)

[B] Frequency-driven Imperceptible Adversarial Attack on Semantic Similarity

[C] Low Frequency Adversarial Perturbation

[D] Harmonic Adversarial Attack Method

**Strengths And Weaknesses:**

Strengths
=======
* The insights provided by the authors in this work are interesting
* The methodology is sound
* The authors provide an ablation and answer a series of interesting questions in the discussion (e.g. whether linear probing is also best for ImageNet 21K

Weaknesses
=========
* While the authors claim ``PLP outperforms the previous state-of-the-art (Ming et al., 2022)'', they miss recent work such as [A]
* While it is possible that any state of the art adversarial attack for computer vision would have sufficed to show the vulnerability of the OOD detector, the authors decided to introduce a new method without properly quantifying its performance nor comparing it to previous SoTA.

[A] Wang, Hualiang, et al. "Clipn for zero-shot ood detection: Teaching clip to say no." Proceedings of the IEEE/CVF International Conference on Computer Vision. 2023.

---

> ### Author Response · Authors · 2024-02-13
> **Reply to Reviewer a6kb**
>
> Thank you for the insightful suggestions and feedback. Below, we aim to address the reviewer’s concerns and remarks.
>
> # Weakness 2 and Requested Changes 1: Adversarial method for OOD detection.
> We were not aware that the mentioned approaches existed before, and we thank the reviewer for pointing them out. Our rationale was that classification-based adversarial attacks could not be applied as they are for OOD detection [1], so we introduced a simple method to verify the OOD adversarial robustness of CLIP. In the revised version, we will clearly state that the method used is not novel nor state-of-the-art, and we will only introduce it to expose the vulnerability of CLIP as an OOD detector.
>
> # Weakness 1 and requested changes 2: CLIPN and state-of-the-art claim of PLP.
>
> CLIPN [2] was published shortly before our submission, so we missed it. We thank the reviewer for indicating this work. In the revised version, we plan to add it to the related work and discuss the key differences.
>
> Briefly, our method does not require designing hand-crafted prompts, does not modify the backbone parameters, nor trains a separate encoder, which is highly costly on large scales. For this reason, CLIPN is only applied to ViT-B/16 and ViT-B/32, and it is not shown whether it scales to larger datasets in the CLIPN [2]. More importantly, CLIPN requires an additional labeled image-text dataset to train the additional text encoder to leverage the “no” prompts. The authors used 3 million images. PLP is shown to scale seamlessly to billion scale parameters (ViT-G) and requires minutes to train a linear layer on ImageNet using the pre-computed representations. We will clarify that our work does not aim to modify the parameters of CLIP or train an auxiliary image/text encoder, and the state-of-the-art results are claimed compared to training-free approaches.
>
> Although it is difficult to generate a 100% fair comparison to CLIPN, we provide a Table in the supplementary with CLIPN, MCM, and PLP (ours) using the CLIP ViT-B/16 trained on LAION-2B.
>
> | Dataset	| iNaturalist | iNaturalist | Texture | Texture | SUN | SUN | Places |Places |
> |--|-|-|-|-|-|-|---|--|
> |    	| FPR95↓  	| AUROC↑ 	| FPR95↓  | AUROC↑  | FPR95↓ | AUROC↑ | FPR95↓ | AUROC↑ |
> | Pseudo-MSP | 74.56   	| 86.88  	| 58.04   | 86.67   | 56.47  | 89.10  | 61.88  | 86.05  |
> | PLP+Energy | 61.41   	| 92.05  	| 45.65   | 88.94   | 57.31  | 88.55  | 59.38  | 85.99  |
> | **Methods that use auxiliary image-text data and a separate text encoder**|
> | CLIPN-A	| 23.94   	| 95.27  	| 40.83   | 90.93   | 26.17  | 93.93  | 33.45  | 92.28  |
>
> We would be interested in whether CLIPN can work with PLP (our method), but no weights are publicly available to investigate this further, and thus, we leave this to future work. Finally, our work systematically studies all 3 large-scale OOD scenarios (unsupervised, label names available, supervised) using CLIP.
>
>
> # Similar curve compared to the original CLIP paper
>
> Our figure that demonstrates the superiority of CLIP across 4 OOD benchmarks is indeed related to Fig 12 and Fig 13 in the original CLIP paper [3]. We highlight the key differences below compared to Radford et al [3]. First, the authors only investigate classification transfer performance (using linear probing) while we investigate unsupervised training-free OOD detection on Imagenet (CLIP Fig.12). In ImageNet OOD detection benchmarks, the supervised ImageNet-pre-trained models have a stronger advantage compared to transfer learning and still CLIP ViT-G outperforms the existing supervised ImageNet models. Second, the authors benchmark natural distribution shift (Fig 13, also known as OOD generalization) using zero-shot, which requires label names. In summary, Fig 12 and 13 of CLIP either require training (linear probing) or access to class names (for zero-shot classification) while we emphasize that CLIP is the best performing unsupervised OOD detection even without label names, and we establish a linear correlation between in-distribution classification accuracy and unsupervised OOD performance only for CLIP models, unlike Fort et al. [4] Figure 10 ( Appendix C.1 Scaling of [supervised] OOD performance with in-distribution test accuracy).
>
> # References
>
> [1] Mao, C., Geng, S., Yang, J., Wang, X., & Vondrick, C. (2022). Understanding zero-shot adversarial robustness for large-scale models.
>
> [2] Wang, H., Li, Y., Yao, H., & Li, X. (2023). Clipn for zero-shot ood detection: Teaching clip to say no. In Proceedings of the IEEE/CVF International Conference on Computer Vision
>
> [3] Radford, A., Kim, J. W., Hallacy, C., Ramesh, A., Goh, G., Agarwal, S., ... & Sutskever, I. (2021, July). Learning transferable visual models from natural language supervision. In International conference on machine learning
>
> [4] Fort, S., Ren, J., & Lakshminarayanan, B. (2021). Exploring the limits of out-of-distribution detection. Advances in Neural Information Processing Systems

---

### Author Response · Authors · 2024-02-14
**Rebuttal submitted**

We would like to thank the reviewers and the TMLR editors for providing us with some additional time to respond to the reviews. We believe we have addressed the majority of the concerns and the requested changes in the revised version of the manuscript. We tried our best to include the requested changes in the reply when possible. Let us know if you need further clarifications or additional experiments.

---

### Decision · Action_Editor_YxSx · 2024-02-29

**Recommendation:** Accept as is

**Comment:**

The following concerns were also raised by reviewers and addressed to their satisfaction by the authors' response and the updated submission:

* The presentation of PLP in Section 3.3 is not accessible to readers with no prior knowledge of related work.
* The differences between PLP and the Fort et al. (2021) and Huang & Li (2021) baselines need clarification.
* The submission should present results on the trainable parameters / performance trade-off.
* The placement of Figure 1 hurts readability.

**Audience:**

All reviewers expressed interest in the results presented in the submission, listing strengths like

* interesting insights (a6kb),
* the submission studies an important question (V5a6),
* the proposed way to curate adversarial examples is interesting (FVMe), and
* the submission introduces new ideas to adapt CLIP representations for OOD detection and to adversarially manipulate OOD images (V5a6).

Reviewer V5a6 notes that the investigation of the observed results is somewhat superficial (in that it misses out on the opportunity to better understand what makes CLIP representations so useful for OOD detection). Reviewer a6kb shares this concern. The authors acknowledge the room for improvement but argue that the investigation is beyond the scope of their paper.

From my vantage point, the observations presented in the submission are interesting to at least part of the TMLR audience, and the submission therefore meets the bar for that aspect of TMLR's acceptance criteria.

**Claims And Evidence:**

Reviewers note the following strengths:

* the methodology is sound (a6kb),
* the ablation experiments answer a series of interesting questions in the discussion (a6kb), and
* there is a strong positive correlation between in-distribution accuracy and OOD detection across multiple benchmarks (V5a6),

all of which go towards supporting the main claims made in the submission. They raised the following concerns:

* The claim that "PLP outperforms the previous state-of-the-art (Ming et al., 2022)" misses out on Wang, Hualiang et al. (2023) (a6kb). The authors respond by explaining the differences between their approach and CLIPN and introducing a comparison in the supplementary material.
* The submission is missing a comparison between the adversarial approach introduced and other adversarial methods (a6kb). In light of this, the reviewer is not convinced by the argument that OOD detection with CLIP is vulnerable to adversarial attacks. The authors amended the paper to clarify that they are not claiming that the method is novel or SOTA and to reduce the scope of the claims supported by the experiment.
* Reviewer FVMe is not convinced that the comparison against the baselines is fair and suggests that the authors should try using the same HuggingFace weights as Ming et al. (2022) and compare Finetune+MSP vs Probing+MSP and Finetune+Energy vs Probing Energy. The authors respond that  they did try to reproduce the results from Ming et al. (2022) and contacted the authors for further clarifications. They also mention that Ming et al. (2022)'s GitHub repository shows that others are facing similar issues. They provide a more detailed analysis in the supplementary material and also provide Finetune+Energy results and clarify the rationale for which methods are compared against each other.

Reviewers are satisfied with the authors' response, and ultimately they all agree that the submission meets the bar in terms of claims and evidence.